# CONT-GRU: FULLY CONTINUOUS GATED RECURRENT UNITS FOR IRREGULAR TIME SERIES

## ABSTRACT

For a long time, RNN-based models, such as RNNs, LSTMs and GRUs, have been used to process time series. However, RNN-based models do not fit well with sporadically (or irregularly) observed real-world data. To this end, some methods *partially* continuously model RNNs/GRUs using ordinary differential equations (ODEs). In this paper, however, we propose Cont-GRU, which models GRUs as delay differential equations (DDEs). By redefining GRUs as DDEs, we show that i) all the parts of GRUs (the hidden state, the reset gate, the update gate, and the update vector) can be interpreted *fully* continuously, and ii) our method does not inherit the limitations of ODEs. In our experiments using 5 real-world datasets and 17 baselines, Cont-GRU outperforms all baselines by non-trivial margins.

## 1 INTRODUCTION

Real-world time series datasets are frequently irregular since some observations can be missing (due to malfunctioning sensors and/or communication channels) and/or observations are collected in an event-driven manner (Brockwell & Davis, 2002; Shumway et al., 2000). However, recurrent neural networks (RNNs), such as long short-term memory (LSTMs (Hochreiter & Schmidhuber, 1997)) and gated recurrent units (GRUs (Cho et al., 2014)), are limited in processing irregular time series. In Figure 1 (a), for instance, GRUs process regularly sampled time series in a discrete manner.

To this end, there are various enhancements for GRUs/RNNs, e.g., GRU-ODE-Bayes, ODE-RNN, NJODE, and so on (Herrera et al., 2021; Lukoševičius & Uselis, 2022; Schirmer et al., 2021; Brouwer et al., 2019; Rusch et al., 2022). These models make fundamental and unique contributions in continuously generalizing GRUs and therefore, they have a strong point in processing irregular time series. ODE-RNN, GRU-ODE-Bayes, and others have used the neural ordinary differential equation (NODE)-based technology to process irregular time series $\{(\mathbf{x}_i, t_i)\}_{i=0}^{N-1}$, where the inter-arrival time $t_i - t_{i-1}$ is not fixed. However, these approaches have, in general, the following limitations:

1. Only the hidden state is continuous on time and other gates are still discrete, which is a half-way continuous generalization of GRUs (cf. Figure 1 (b)).
2. There exists a discontinuity in reading $\mathbf{x}_i$ at time $t_i$ — the discontinuity point is called as *jump* since the hidden state $\mathbf{h}(t_i)$ jumps to a different location $\mathbf{h}'(t_i) = j(\mathbf{h}(t_i), \mathbf{x}_i; \boldsymbol{\theta}_j)$ suddenly. They need an auxiliary neural network $j$ to perform the jump operation.
3. The trajectory from $\mathbf{h}'(t_{i-1})$ to $\mathbf{h}(t_i)$ is modeled by a NODE and determined only by $\mathbf{h}'(t_{i-1})$ (and other gates at time $t_{i-1}$).
4. Moreover, the topologies of $\mathbf{h}'(t_{i-1})$ and $\mathbf{h}(t_i)$ are identical — simply speaking, large updates are not made from $\mathbf{h}'(t_{i-1})$ to $\mathbf{h}(t_i)$. In fact, this is the well-known *homeomorphic* limitation of NODEs. There exists a countermeasure for this limitation (Dupont et al., 2019). However, this countermeasure is not as effective as our method since it does not make substantial changes to NODEs.

In this paper, we redefine GRUs as delay differential equations (DDEs) that reflect past observations to the current hidden state for processing irregular time series and fully continuously generalize GRUs (cf. Figure 1 (c)). Our method does not have the aforementioned limitations of NODE-based methods. First, we consider the following general form of GRUs, including the reset gate, the update

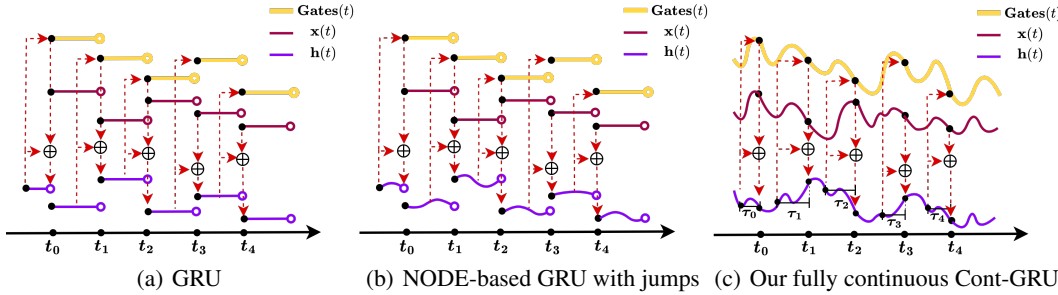

Figure 1: Existing methods vs. Cont-GRU. 'Gates' means the various gates of GRUs. In the first two methods, those gates are not continuous.

gate, and the update vector:

$$
\begin{aligned}
\mathbf{h}(t) &:= \mathbf{z}(t) \odot \mathbf{h}(t - \tau) + (1 - \mathbf{z}(t)) \odot \mathbf{g}(t), \\
\mathbf{z}(t) &:= \sigma\big(\mathbf{W}_z \mathbf{x}(t) + \mathbf{U}_z \mathbf{h}(t - \tau) + \mathbf{b}_z\big) \\
\mathbf{g}(t) &:= \phi\big(\mathbf{W}_g \mathbf{x}(t) + \mathbf{U}_g\big(\mathbf{r}(t) \odot \mathbf{h}(t - \tau)\big) + \mathbf{b}_g\big), \\
\mathbf{r}(t) &:= \sigma\big(\mathbf{W}_r \mathbf{x}(t) + \mathbf{U}_r \mathbf{h}(t - \tau) + \mathbf{b}_r\big),
\end{aligned}
\tag{1}
$$

where $\tau > 0$ is a delay factor. In our continuous GRU regime, $\tau$ adaptively changes for a downstream task (see the discussion in Section. 3.3).

We later calculate the time derivative terms of $\mathbf{h}(t)$, $\mathbf{z}(t)$, $\mathbf{g}(t)$, and $\mathbf{r}(t)$ in Section. 3.2 to convert the general form into a DDE. To our knowledge, our model, called ***Continuous GRU*** (Cont-GRU), is the first fully continuous interpretation of GRUs. Our method can be summarized as follows:

1. We calculate the time derivatives of the hidden state $\mathbf{h}(t)$, the reset gate $\mathbf{r}(t)$, the update gate $\mathbf{z}(t)$, and the update vector $\mathbf{g}(t)$ of GRUs in Section. 3.2.

2. We then define an augmented delay differential equation (DDE) in Eq. equation 6 after combining all those time derivative terms. The advantages of our model are as follows:

   (a) DDEs share the same base philosophy as that of GRUs, which is past information influences current output. DDEs are for modeling these time-delay systems. In particular, the delay factor $\tau$ dynamically changes over time whereas $\tau = 1$ in the original GRU design.

   (b) Using an interpolation algorithm, we convert the discrete time series sample $\{(\mathbf{x}_i, t_i)\}_{i=0}^{N-1}$ into a continuous path $\mathbf{x}(t)$, where $\mathbf{x}(t_i) = (\mathbf{x}_i, t_i)$ at each observation time point $t_i$ and for other non-observed time points, the interpolation algorithm fills out appropriate values.

   (c) There is no need to use the jump mechanism, and Cont-GRU defines all the gates and hidden state continuously over time.

   (d) Our DDE-based method does not have the limitation of ODEs — DDEs are not homeomorphic — and therefore, we expect better representation learning capability which is important for downstream tasks. In addition, our Cont-GRU model is relatively lightweight in terms of memory footprint. See Appendix A for more discussion.

## 2 RELATED WORK

**Continuous-time time series processing**  Deep learning models based on differential equations are commonly utilized for processing time series. Many of them rely on a technology called neural ordinary differential equations (NODEs (Chen et al., 2018)), which solve the following initial value problem:

$$
\mathbf{h}(t_i) = \mathbf{h}(t_{i-1}) + \int_{t_{i-1}}^{t_i} f(t, \mathbf{h}(t), \boldsymbol{\theta}_f) dt,
\tag{2}
$$

where $f$, called *ODE function*, is a neural network which is parameterized by $\boldsymbol{\theta}_f$ and approximates $\frac{d\mathbf{h}(t)}{dt}$. We can get $\mathbf{h}(t_i)$ by solving the initial value problem with various ODE solvers. However, NODEs are homeomorphic. In other words, the mapping from $\mathbf{h}(t_{i-1})$ to $\mathbf{h}(t_i)$ continuously changes in a bijective manner, which is too restrictive in some cases for complicated tasks and therefore, augmenting $\mathbf{h}(t)$ with zeros had been proposed in (Dupont et al., 2019). We also consider this augmentation technique to enhance some baselines for thorough experiments. However, this increases the model size and computation amount.

NODEs utilize Eq. equation 2 between two observations to model the evolutionary process of the hidden state $\mathbf{h}(t)$ in a continuous time domain. Following this idea, various differential equation-based time series models have been recently proposed. Most of them use a mechanism called *jump*. A jump means that i) the hidden state jumps to a different location after reading a new observation, aided by an auxiliary jump network and ii) a new initial value problem should be solved (cf. Figure 1 (b)) — in other words, this processing paradigm is able to process irregular time series since the jumping can happen anytime. ODE-RNN (Rubanova et al., 2019), GRU-ODE-Bayes (Brouwer et al., 2019), and neural jump ODEs (NJODEs) (Herrera et al., 2021) are representative methods for this approach.

In (Kidger et al., 2020), neural controlled differential equations (NCDEs) were proposed to process irregular time series by using the controlled differential equation paradigm. Since NCDEs create a continuous path by using interpolation methods, they can continuously generalize the hidden state without any jump mechanism. In other words, NCDEs read a continuous path and evolve the hidden state $\mathbf{h}(t)$ continuously over time.

**Delay differential equations**  Delay differential equations (DDEs) are a type of differential equation in mathematics that uses the value of a function from a previous time to determine the derivative of the function at a given time. Considering the characteristics of DDEs, a time-delay RNN model suitable for temporal correlation and volatile financial time series was also proposed in (Kim, 1998). DDEs overcome the limitations of NODEs. In particular, it overcomes the limitations of NODEs well in physical or physiological systems where the effect of time delay cannot be avoided. In this context, two interesting papers (Zhu et al., 2021; 2022) have been published. In those papers, the simplest form of neural delay differential equations (NDDEs) can be written as follows:

$$\mathbf{h}(t_i) = \mathbf{h}(t_{i-1}) + \int_{t_{i-1}}^{t_i} f(t, \mathbf{h}(t), \mathbf{h}(t - \tau), \boldsymbol{\theta}_f)dt, \tag{3}$$

where $\tau$ is a delay effect. The difference between Eq. equation 2 and Eq. equation 3 is that NDDEs consider the hidden state of past times.

## 3 PROPOSED METHOD

In this section, we describe our proposed fully continuous GRU concept. First, we redefine GRUs as DDEs. Applying the DDE to the GRU is particularly suitable for capturing time-dependent correlations and predicting for time series. Unlike existing RNN-based models where the time-delay factor $\tau$ is fixed to 1, our proposed model, Cont-GRU, has the flexibility to dynamically control the delay term $\tau$. After redefining GRUs as DDEs, Cont-GRU generalizes all the gates, including the update gate, the update vector and the reset gate, continuously. Generalizing all the gates continuously enables accurate understanding the flow of time series when performing downstream tasks, especially for time series classification and forecasting. In addition, we do not need an jump network.

### 3.1 OVERALL WORKFLOW

Figure 1 (c) shows the overall workflow diagram of our method, Cont-GRU, which is defined as follows:

1. A continuous path $\mathbf{x}(t)$ is created from a discrete time series sample by an interpolation algorithm — one can choose any interpolation method, e.g., natural cubic spline.
2. All the reset gate $\mathbf{r}(t)$, the update gate $\mathbf{z}(t)$, the update vector $\mathbf{g}(t)$, and the hidden state $\mathbf{h}(t)$ of GRUs are modeled as an augmented DDE of Eq. equation 6, which means that they are all continuous in our framework.

3. After that, there is one more fully connected layer to further process $\mathbf{h}(t)$ for a downstream task, i.e., output layer.

## 3.2 FULLY CONTINUOUS GRUS

In order to continuously generalize GRUs, we need to calculate the time derivatives of GRU's various parts. Considering Eq. equation 1, we can define them as an augmented DDE in Eq. equation 6.

**Time derivative of $\mathbf{h}(t)$** Since the hidden state $\mathbf{h}(t)$ is a composite function of $\mathbf{r}(t)$, $\mathbf{z}(t)$, and $\mathbf{g}(t)$, the derivative of $\mathbf{h}(t)$ can be written as follows:

$$
\begin{aligned}
\frac{d\mathbf{h}(t)}{dt} &= \frac{d\mathbf{z}(t)}{dt} \odot \mathbf{h}(t-\tau) + \mathbf{z}(t) \odot \frac{d\mathbf{h}(t-\tau)}{dt} - \frac{d\mathbf{z}(t)}{dt} \odot \mathbf{g}(t) + (1-\mathbf{z}(t)) \odot \frac{d\mathbf{g}(t)}{dt}, \\
&= \frac{d\mathbf{z}(t)}{dt} \odot \big(\mathbf{h}(t-\tau) - \mathbf{g}(t)\big) + \mathbf{z}(t) \odot \big(\frac{d\mathbf{h}(t-\tau)}{dt} - \frac{d\mathbf{g}(t)}{dt}\big) + \frac{d\mathbf{g}(t)}{dt}, \\
&= \frac{d\mathbf{z}(t)}{dt} \odot \zeta(t, t-\tau) + \mathbf{z}(t) \odot \frac{d\zeta(t, t-\tau)}{dt} + \frac{d\mathbf{g}(t)}{dt},
\end{aligned}
\tag{4}
$$

where $\zeta(t, t-\tau) = \mathbf{h}(t-\tau) - \mathbf{g}(t)$. So, we can write $\frac{d\mathbf{h}(t)}{dt}$ as follows:

$$
\frac{d\mathbf{h}(t)}{dt} = \frac{d(\mathbf{z}(t) \odot \zeta(t, t-\tau))}{dt} + \frac{d\mathbf{g}(t)}{dt}.
\tag{5}
$$

Other derivatives for $\mathbf{z}(t)$, $\mathbf{g}(t)$, and $\mathbf{r}(t)$ are in Appendix I. Finally, the time derivatives of $\mathbf{h}(t), \mathbf{r}(t), \mathbf{z}(t)$, and $\mathbf{g}(t)$ is written as follows :

$$
\frac{d}{dt}
\begin{bmatrix}
\mathbf{h}(t) \\
\mathbf{z}(t) \\
\mathbf{g}(t) \\
\mathbf{r}(t)
\end{bmatrix}
:=
\begin{bmatrix}
\frac{d(\mathbf{z}(t) \odot \zeta(t, t-\tau))}{dt} + \frac{d\mathbf{g}(t)}{dt} \\
\sigma\big(\mathbf{A}(t, t-\tau)\big)(1 - \sigma(\mathbf{A}(t, t-\tau)))\frac{d\mathbf{A}(t, t-\tau)}{dt} \\
\big(1 - \phi^2(\mathbf{B}(t, t-\tau))\big)\frac{d\mathbf{B}(t, t-\tau)}{dt} \\
\sigma\big(\mathbf{C}(t, t-\tau)\big)(1 - \sigma(\mathbf{C}(t, t-\tau)))\frac{d\mathbf{C}(t, t-\tau)}{dt}
\end{bmatrix}.
\tag{6}
$$

We note that the above definition becomes a DDE since $\zeta, \mathbf{A}, \mathbf{B}$, and $\mathbf{C}$ have internally $\mathbf{h}(t-\tau)$. $\frac{d\mathbf{x}(t)}{dt}$ contained by the derivatives of $\mathbf{A}, \mathbf{B}$, and $\mathbf{C}$ can also be calculated since we use an interpolation method to construct $\mathbf{x}(t)$ (see Section 4.5).

## 3.3 TRAINING METHOD

In Alg. 1, we show our training algorithm. Since our Cont-GRU can be used for various tasks, we show a brief pseudo-code of the training method in Alg. 1. For a more concrete example, suppose a time series classification task with $(\{(\mathbf{x}_i, t_i)\}_{i=0}^{N-1}, \mathbf{y})$, where $\mathbf{y}$ is the ground-truth class label of the discrete time series sample. For this, we first solve the following integral problem:

$$
\begin{bmatrix}
\mathbf{h}(t_{N-1}) \\
\mathbf{r}(t_{N-1}) \\
\mathbf{z}(t_{N-1}) \\
\mathbf{g}(t_{N-1})
\end{bmatrix}
=
\begin{bmatrix}
\mathbf{h}(0) \\
\mathbf{r}(0) \\
\mathbf{z}(0) \\
\mathbf{g}(0)
\end{bmatrix}
+
\int_0^{t_{N-1}} \frac{d}{dt}
\begin{bmatrix}
\mathbf{h}(t) \\
\mathbf{r}(t) \\
\mathbf{z}(t) \\
\mathbf{g}(t)
\end{bmatrix} dt,
\tag{7}
$$

where $\mathbf{h}(0), \mathbf{r}(0), \mathbf{z}(0), \mathbf{g}(0)$ are set in the same way as the original discrete GRU.

We then feed $\mathbf{h}(t_{N-1})$ into a following output layer with a softmax activation to predict its class label $\hat{\mathbf{y}}$, where the task loss $L$ is a cross-entropy loss between the prediction $\hat{\mathbf{y}}$ and the ground-truth $\mathbf{y}$. During the process, one can easily calculate the gradients using either the standard backpropagation or the adjoint sensitivity method (Chen et al., 2018).

**Adaptive delay factor** In the perspective of solving Eq. equation 7, we can use an ODE solver that progressively updates the augmented state of $[\mathbf{h}(t), \mathbf{r}(t), \mathbf{z}(t), \mathbf{g}(t)]$ from $t = 0$ to $t_{N-1}$ by referring to its time derivative term in Eq. equation 6. During the solving process, we found that $\mathbf{h}(t-\tau)$ can be approximated by $\mathbf{h}(t-s)$, where $s$ is an adaptive step size of DOPRI, a default ODE solver in many papers. DOPRI internally estimates a step error and determines the step-size $s$ adaptively every

step (Dormand & Prince, 1980). In principle, the step error depends on the degree of *volatility* of the learned DDE. In other words, it increases $s$ when the augmented state of $[\mathbf{h}(t), \mathbf{r}(t), \mathbf{z}(t), \mathbf{g}(t)]$ does not change notably from $t$ to $t + s$, i.e., non-volatile DDEs. If not, it decreases $s$. For complicated downstream tasks, complicated DDEs should be learned and the volatility increases. Therefore, one can say that the adaptive step size $s$ varies for a target downstream task.

---

**Algorithm 1:** How to train Cont-GRU

---

**Input:** Training data $D_{train}$ Validating data $D_{val}$,
     Maximum iteration numbers $max\_iter$
1 Initialize the parameters $\boldsymbol{\theta}$, e.g., $\mathbf{W}_h$, $\mathbf{U}_h$, etc.;
2 Create a continuous path $\mathbf{x}(t)$ for each $\{(\mathbf{x}_i, t_i)\}_{i=0}^{N-1}$;
3 $i \leftarrow 0$; **while** $i < max\_iter$ **do**
4     Train $\boldsymbol{\theta}$ and using a task loss $L$;
5     Validate and update the best parameters $\boldsymbol{\theta}^*$ with
         $D_{val}$;
6     $i \leftarrow i + 1$;
7 **return** $\boldsymbol{\theta}^*$;

---

**Well-posedness** The well-posedness[1] of NDDEs was already proved in (Lyons et al., 2004, Theorem 1.3) under the mild condition of the Lipschitz continuity. We show that our fully continuous GRUs are also well-posed. Almost all activations, such as ReLU, Leaky ReLU, Tanh, Sigmoid, ArcTan, and Softsign, have a Lipschitz constant of 1. Other common neural network layers, such as dropout, batch normalization, and other pooling methods, have explicit Lipschitz constant values. Therefore, the Lipschitz continuity of

$\frac{d\mathbf{h}(t)}{dt}$, $\frac{d\mathbf{r}(t)}{dt}$, $\frac{d\mathbf{z}(t)}{dt}$, and $\frac{d\mathbf{g}(t)}{dt}$ can be fulfilled in our case. Accordingly, it is a well-posed problem. Thus, its training process is stable in practice.

### 3.4 DISCUSSION

In Figure 1, we compare our proposed method with GRU and GRU-ODE-Bayes, a famous jump-based method. First, our method does not need to use the jump mechanism since i) the time derivative of $\mathbf{x}(t)$ can be properly defined, and ii) our augmented DDE definition keeps reading the time derivative. Second, our continuous generalization makes sense mathematically since we consider the time derivative terms of the reset gate, the update gate, and the update vector in conjunction with the time derivative term of the hidden state. In fact, calculating the time derivative of the hidden state requires the time derivatives of other gates, which were ignored in GRU-ODE-Bayes. In addition, our DDE-based fully continuous GRUs do not have the homeomorphic limitation of ODEs.

Owing to these facts of Cont-GRU, it shows more robust processing for irregular time series. In our experiments, we compare our method with existing methods in diverse environments.

## 4 EXPERIMENTS

In this section, we describe our experimental environments and results. We conduct experiments with time series classification and forecasting. We repeat training and testing procedures with five different random seeds and report their mean and standard deviation scores.

**Experimental environments** We list all the hyperparameter settings and our 17 baselines in Appendix B, D and C. We focus on accuracy in the main paper and all memory usage and runtime are reported in Appendix M.

### 4.1 FORECAST WEATHER IN VARIOUS SEQUENCE LENGTHS

Ecosystems and other social systems have long been accustomed to predictable weather characteristics. Unexpected weather conditions, such as global warming or extreme weather, occur frequently from recently. Therefore, predicting these future weather conditions is very important to society (Salman et al., 2015; Grover et al., 2015). Due to the nature of weather data, it is challenging to predict long-distance weather conditions, but it is an important issue for our society.

In this paper, we forecast weather conditions with various sequence lengths. We use the United State Historical Climatology Network (USHCN) daily dataset (Menne & Williams Jr, 2009). USHCN

---

[1] A well-posed problem means i) its solution uniquely exists, and ii) its solution continuously changes as input data changes.

data includes five climatic variables (daily temperatures, precipitation, snow, and so on) for 1,218 meteorological stations across the United States over 150 years. We use a subset of 1,114 meteorological stations over four years from 1996 to 2000 using the cleaning method proposed in (Brouwer et al., 2019).

Table 1: USHCN

| Model | Test MSE | | |
|---|---|---|---|
| | 16 sequence | 32 sequence | 64 sequence |
| RNN | 0.25 ± 0.00 | 0.24 ± 0.04 | 0.24 ± 0.06 |
| LSTM | 0.28 ± 0.01 | 0.27 ± 0.00 | 0.26 ± 0.03 |
| GRU | 0.26 ± 0.01 | 0.25 ± 0.02 | 0.24 ± 0.02 |
| NODE | 0.23 ± 0.02 | 0.25 ± 0.04 | 0.22 ± 0.05 |
| ODE-RNN | 0.21 ± 0.03 | 0.23 ± 0.05 | 0.25 ± 0.07 |
| GRU-$\Delta t$ | 0.28 ± 0.04 | 0.30 ± 0.03 | 0.24 ± 0.06 |
| GRU-D | 0.30 ± 0.02 | 0.30 ± 0.04 | 0.30 ± 0.04 |
| GRU-ODE | 0.27 ± 0.07 | 0.31 ± 0.07 | 0.31 ± 0.02 |
| Latent-ODE | 0.34 ± 0.00 | 0.38 ± 0.00 | 0.36 ± 0.01 |
| Augmented-ODE | 0.33 ± 0.01 | 0.35 ± 0.02 | 0.33 ± 0.02 |
| ACE-NODE | 0.35 ± 0.09 | 0.38 ± 0.05 | 0.32 ± 0.04 |
| GRU-ODE-Bayes | 0.39 ± 0.08 | 0.42 ± 0.01 | 0.47 ± 0.01 |
| NJODE | 0.37 ± 0.06 | 0.39 ± 0.03 | 0.48 ± 0.06 |
| NCDE | 0.24 ± 0.08 | 0.41 ± 0.09 | 0.39 ± 0.01 |
| ANCDE | 0.22 ± 0.04 | 0.30 ± 0.07 | 0.32 ± 0.02 |
| EXIT | 0.28 ± 0.01 | 0.27 ± 0.00 | 0.27 ± 0.01 |
| SCINet | 0.15 ± 0.01 | 0.11 ± 0.00 | 0.28 ± 0.01 |
| **Cont-GRU** | **0.06 ± 0.00** | **0.09 ± 0.01** | **0.18 ± 0.02** |

**Experimental results**  Table 1 shows one of the most extensively used benchmark experiments, i.e., time series forecasting with the USHCN weather dataset. To create a challenging task, we evaluate various experimental settings. We conduct experiments after reading 128 sequences for forecasting the next 16, 32, and 64 sequences — GRU-ODE-Bayes forecasts up to the next 3 sequences and our settings are much more challenging. Cont-GRU shows the best accuracy in various output sequence lengths. Jump-based models, i.e., GRU-ODE-Bayes and NJODE, show poor accuracy, which shows that their piece-wise continuous concepts do not effectively process weather data. Exceptionally, ODE-RNN, whose jump mechanism is based on RNN cells, shows good performance. Since GRU-based models are typically used for time series forecasting, some of them show reasonable results with small standard deviations. We visualize the prediction results of Cont-GRU and the top-2 baseline models for 32 sequences in Appendix E.

### 4.2 PREDICT PATIENT CONDITIONS WITH HIGHLY IRREGULAR TIME SERIES

Table 2: PhysioNet Sepsis

| Model | AUROC |
|---|---|
| NODE | 0.53 ± 0.04 |
| ODE-RNN | 0.87 ± 0.02 |
| GRU-$\Delta t$ | 0.88 ± 0.01 |
| GRU-D | 0.87 ± 0.02 |
| GRU-ODE | 0.85 ± 0.01 |
| Latent-ODE | 0.79 ± 0.01 |
| Augmented-ODE | 0.83 ± 0.02 |
| ACE-NODE | 0.80 ± 0.01 |
| GRU-ODE-Bayes | 0.52 ± 0.01 |
| NJODE | 0.53 ± 0.01 |
| NCDE | 0.88 ± 0.01 |
| ANCDE | 0.90 ± 0.00 |
| EXIT | 0.91 ± 0.00 |
| **Cont-GRU** | **0.93 ± 0.04** |

Sepsis (Reyna et al., 2019; Reiter, 2005) is a life-threatening condition caused by bacteria or bacterial toxins in the blood. About 1.7 million people develop sepsis in the U.S., and 270,000 die from sepsis in a year. More than a third of people who die in U.S. hospitals have sepsis. Early sepsis prediction could potentially save lives, so this experiment is especially meaningful. The dataset used in this paper consists of data from 40,335 patients in intensive care units (ICU). The data consists of 5 static variables that do not change over time, such as gender and age, and 34 non-static variables, such as the respiratory rate or partial pressure of carbon dioxide from arterial blood (PaCO2). This data can be described as an irregular time series dataset with 90% of values removed from the original full data to protect the privacy of patients. To classify the onset of sepsis, we consider the first 72 hours of the patient's hospitalization.

**Experimental results**  Table 2 shows our experimental results of the time series classification with PhysioNet Sepsis. We conduct the time series classification task with observation intensity (OI) as an additional variable, which was used in (Kidger et al., 2020). We report AUROC rather than accuracy because the dataset is significantly imbalanced. Cont-GRU shows the best performance and the model size is small in comparison with other differential equation-based models. In this dataset, however, more than 90% of values are missing to protect the privacy of patients. For this reason, GRU-ODE-Bayes and NJODE do not show good performance. They can process irregular time series, but their accuracies are worse than others. We consider that this is because they piece-wise continuously generalize the hidden state only. However, all NCDE-based models, i.e., NCDE, ANCDE, and EXIT, show reasonable results since they fully continuously generalize the hidden state.

## 4.3 FORECAST VOLATILE STOCK PRICES AND VOLUMES

Table 3: Google Stock

| Model | Test MSE | | |
| --- | --- | --- | --- |
| | 30% dropped | 50% dropped | 70% dropped |
| NODE | 0.057 ± 0.006 | 0.054 ± 0.005 | 0.052 ± 0.013 |
| ODE-RNN | 0.116 ± 0.018 | 0.145 ± 0.006 | 0.129 ± 0.011 |
| GRU-$\Delta t$ | 0.145 ± 0.002 | 0.146 ± 0.001 | 0.145 ± 0.002 |
| GRU-D | 0.143 ± 0.002 | 0.145 ± 0.002 | 0.146 ± 0.002 |
| GRU-ODE | 0.064 ± 0.009 | 0.057 ± 0.003 | 0.059 ± 0.004 |
| Latent-ODE | 0.052 ± 0.005 | 0.053 ± 0.001 | 0.054 ± 0.007 |
| Augmented-ODE | 0.045 ± 0.004 | 0.051 ± 0.005 | 0.057 ± 0.002 |
| ACE-NODE | 0.044 ± 0.002 | 0.053 ± 0.008 | 0.056 ± 0.003 |
| GRU-ODE-Bayes | 0.175 ± 0.001 | 0.185 ± 0.022 | 0.197 ± 0.013 |
| NJODE | 0.185 ± 0.002 | 0.191 ± 0.012 | 0.181 ± 0.031 |
| NCDE | 0.056 ± 0.015 | 0.054 ± 0.002 | 0.056 ± 0.007 |
| ANCDE | 0.048 ± 0.012 | 0.047 ± 0.001 | 0.049 ± 0.004 |
| EXIT | 0.042 ± 0.020 | 0.045 ± 0.001 | 0.046 ± 0.002 |
| SCINet | 0.021 ± 0.004 | 0.027 ± 0.003 | 0.031 ± 0.004 |
| **Cont-GRU** | **0.007 ± 0.001** | **0.006 ± 0.001** | **0.007 ± 0.002** |

Stock prices are the results of the combination of social conditions and people's psychological factors (Andreassen, 1987; Wäneryd, 2001). Thus, accurate stock price forecasting is a very challenging task. Particularly, forecasting stock prices, including the duration of COVID-19, makes our task more challenging and can properly evaluate time series forecasting models. We use the Google Stock data (Alphabet, 2021), which has six variables, i.e., the trading volumes of Google in conjunction with its high, low, open, close, and adjusted closing prices. We use the period from 2011 to 2021 of Google stock data, purposely including the COVID-19 period. The goal is, given the past 20 days of the time series values, to forecast the high, low, open, close, adjusted closing price, and volumes at the very next 10 days.

**Experimental results** The experimental results on Google Stock are in Table 3. We conduct experiments after randomly dropping 30 %, 50%, and 70% of observations in each time series sample. Overall, our model, Cont-GRU, shows the best accuracy. One impressive outcome of our method is that it is not greatly affected by the dropping ratio. ODE-based models, except ODE-RNN, show reasonable results and CDE-based models show better results than ODE-based models. Various visualizations of the forecasting results are in Appendix F.

Figure 2 shows the difference between the reset gates of Cont-GRU and GRU-ODE-Bayes. The role of the reset gate is to determine how much of the previous hidden state is reflected. The red line in Figure 2 shows the stock market price for the 20-day period from April 30 to May 28, 2019. One can see that the reset gate of GRU-ODE-Bayes does not fluctuate much, but the reset gate of Cont-GRU captures meaningful information. More visualizations of values in the reset gate are in Appendix G.

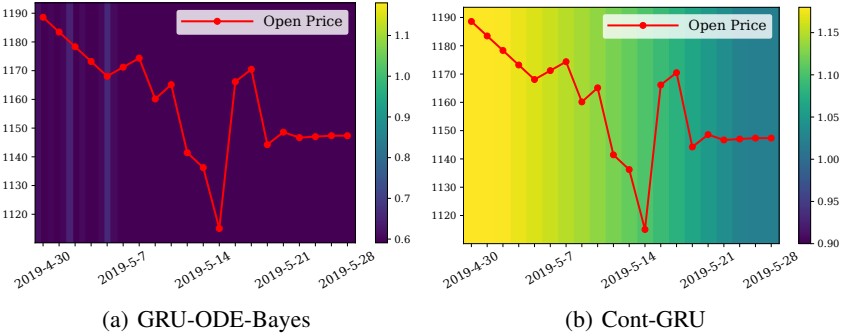

(a) GRU-ODE-Bayes                    (b) Cont-GRU

Figure 2: Values in the reset gate. Our method gives higher weights to recent observations whereas GRU-ODE-Bayes gives almost equal weights to all observations, which shows the correctness of our method.

Table 4: ETT datasets

| Models | ETTh1 | | | ETTh2 | | |
|---|---|---|---|---|---|---|
| Horizons | 24 | 48 | 168 | 24 | 48 | 168 |
| GRU | 0.293 ± 0.011 | 0.355 ± 0.006 | 0.437 ± 0.006 | 0.191 ± 0.002 | 0.224 ± 0.005 | 0.313 ± 0.001 |
| LSTM | 0.304 ± 0.010 | 0.368 ± 0.018 | 0.501 ± 0.018 | 0.212 ± 0.008 | 0.315 ± 0.008 | 0.322 ± 0.004 |
| RNN | 0.334 ± 0.014 | 0.394 ± 0.002 | 0.585 ± 0.002 | 0.199 ± 0.004 | 0.284 ± 0.009 | 0.451 ± 0.003 |
| NODE | 0.440 ± 0.004 | 0.504 ± 0.002 | 0.918 ± 0.008 | 0.110 ± 0.005 | 0.347 ± 0.008 | 0.963 ± 0.008 |
| ODE-RNN | 0.551 ± 0.012 | 0.473 ± 0.014 | 0.554 ± 0.004 | 0.145 ± 0.004 | 0.281 ± 0.003 | 0.336 ± 0.004 |
| GRU-$\Delta t$ | 0.430 ± 0.002 | 0.449 ± 0.004 | 0.576 ± 0.008 | 0.118 ± 0.004 | 0.215 ± 0.008 | 0.271 ± 0.010 |
| GRU-D | 0.439 ± 0.004 | 0.445 ± 0.004 | 0.572 ± 0.007 | 0.107 ± 0.004 | 0.217 ± 0.007 | 0.283 ± 0.012 |
| GRU-ODE | 0.431 ± 0.008 | 0.515 ± 0.004 | 1.241 ± 0.014 | 0.132 ± 0.009 | 0.210 ± 0.007 | 2.863 ± 0.041 |
| Latent-ODE | 0.487 ± 0.006 | 0.510 ± 0.006 | 0.548 ± 0.008 | 0.198 ± 0.004 | 0.204 ± 0.001 | 0.398 ± 0.000 |
| Augmented-ODE | 0.462 ± 0.007 | 0.471 ± 0.011 | 0.580 ± 0.010 | 0.213 ± 0.004 | 0.304 ± 0.002 | 0.441 ± 0.002 |
| ACE-NODE | 0.384 ± 0.009 | 0.409 ± 0.009 | 0.499 ± 0.009 | 0.199 ± 0.006 | 0.301 ± 0.004 | 0.357 ± 0.006 |
| GRU-ODE-Bayes | 0.511 ± 0.011 | 0.527 ± 0.002 | 0.507 ± 0.004 | 0.304 ± 0.012 | 0.441 ± 0.008 | 0.507 ± 0.009 |
| NJODE | 0.600 ± 0.020 | 0.624 ± 0.031 | 0.701 ± 0.018 | 0.327 ± 0.021 | 0.417 ± 0.011 | 0.513 ± 0.013 |
| NCDE | 0.265 ± 0.001 | 0.457 ± 0.002 | 0.522 ± 0.001 | 0.207 ± 0.000 | 0.548 ± 0.001 | 0.744 ± 0.009 |
| ANCDE | 0.257 ± 0.008 | 0.331 ± 0.007 | 0.473 ± 0.009 | 0.187 ± 0.002 | 0.233 ± 0.004 | 0.312 ± 0.006 |
| EXIT | 0.244 ± 0.007 | 0.324 ± 0.006 | 0.481 ± 0.002 | 0.176 ± 0.002 | 0.220 ± 0.006 | 0.322 ± 0.009 |
| SCINet | 0.421 ± 0.004 | 0.368 ± 0.008 | 0.451 ± 0.006 | 0.188 ± 0.007 | 0.279 ± 0.001 | 0.505 ± 0.006 |
| Cont-GRU | **0.220 ± 0.001** | **0.302 ± 0.002** | **0.405 ± 0.004** | **0.092 ± 0.000** | **0.119 ± 0.002** | **0.191 ± 0.004** |

## 4.4 Forecast electricity transformer temperatures with very long time series

The electricity transformer temperature (ETT) plays a crucial role in solving the power distribution problem, which is about distributing electricity to different areas according to their sequential use. However, it is difficult to predict demand at a location since it varies for various factors. Improving the accuracy of predicting future electricity usage is a challenging but important problem as incorrect predictions can damage electrical transformers. Following prior works (Zhou et al., 2021; Liu et al., 2021), we use the two datasets, ETTh1 and ETTh2, sampled every hour. We forecast the next 24/48/168 observations — the input length is the same as the output length in our settings.

**Experimental results** The experimental results on ETT datasets are in Table 4. For ETTh1, NCDE-based models show reasonable results. Conventional time series models, such as GRU, GRU-$\Delta t$, GRU-D, LSTM, RNN, and SCINet, also show reasonable performances across all time sequences. However, jump-based continuous-time models show poor performances in them. For ETTh2, all baselines show reasonable performances for the sequences of 24 and 48. For 168, however, most models perform poorly. In particular, NODE-based models show the worst performance for long sequences. Our model, Cont-GRU, shows the highest accuracy in all cases.

## 4.5 Empirical study on interpolation methods

In this section, we further experiment with several interpolation methods to create a continuous path $\mathbf{x}(t)$ from $\{(\mathbf{x}_i, t_i)\}_{i=0}^{N-1}$ and compare their results. The results are in Table 5. We test with the natural cubic spline (McKinley & Levine, 1998), linear control (Martin et al., 1995), and cubic Hermite spline (De Boor et al., 1987) methods. We show that the interpolation method leads to the continuous derivative of $\frac{d\mathbf{x}(t)}{dt}$ in Appendix H, which enables our DDE-based *fully continuous* Cont-GRU.

**Natural cubic spline** The natural cubic spline method must have access to the entire time series data for this control signal before constructing a continuous path. Changes to one previous data point do not affect the overall structure. This method can be integrated numerically quickly because it is relatively smooth and changes slowly.

**Linear control** The linear control method generates the simplest and most natural control signal among the interpolation methods. An interpolated path is generated while applying the linear in-

terpolation between observations. This linear control defines discrete online control paths for all observed data across all time, and so has the same online qualities as RNNs.

**Cubic Hermite spline**   The linear control method is discrete, which can be a drawback. However, the cubic Hermite spline interpolation method smooths out discontinuities. This method achieves this by combining adjacent observations with a cubic spline. Comparing the cubic Hermite spline method with the natural cubic spline method, the main difference is that equations are solved independently at each point in time.

**Sensitivity to interpolation methods**   For USHCN, the cubic Hermite spline method shows the best performance. However, all interpolation methods show good results. In Google Stock, the linear control method and the cubic Hermite spline method show the best performance among the three interpolation methods. However, all three interpolation results are significantly better than the existing baselines.

Table 5: Interpolation methods

| Interpolation Methods | USHCN | Stock |
|---|---|---|
| Natural Cubic Spline | 0.15 ± 0.03 | **0.006 ± 0.001** |
| Linear Control | 0.16 ± 0.02 | 0.008 ± 0.001 |
| Cubic Hermite Spline | **0.14 ± 0.01** | **0.006 ± 0.001** |

Table 6: Perturbing the hidden state (HP) vs. the continuous data path (DP) in Cont-GRU

| Method | USHCN | Sepsis |
|---|---|---|
| Cont-GRU (DP) | 0.15 ± 0.01 | 0.83 ± 0.02 |
| Cont-GRU (HP) | 0.17 ± 0.01 | 0.77 ± 0.01 |
| Cont-GRU | **0.14 ± 0.01** | **0.93 ± 0.04** |

## 4.6   DDE VS. INTERPOLATION

In order to enable our proposed DDE-based continuous GRUs, we rely on an interpolation method to define the continuous path $\mathbf{x}(t)$. Therefore, we compare the following two variants to know which part more contributes to downstream tasks:

1. We perturb the hidden state using $\mathbf{h}(t) + \boldsymbol{\epsilon}$, where $\boldsymbol{\epsilon} \sim \mathcal{N}(\mathbf{0}, \boldsymbol{\sigma}^2)$ and $\boldsymbol{\sigma}$ is an estimated std. dev. of the hidden state of Cont-GRU. We denote this perturbation as "Cont-GRU (HP)."
2. We perturb the continuous path using $\mathbf{x}(t) + \boldsymbol{\epsilon}$, where $\boldsymbol{\epsilon} \sim \mathcal{N}(\mathbf{0}, \boldsymbol{\sigma}^2)$ and $\boldsymbol{\sigma}$ is an estimated std. dev. of data. We denote this perturbation as "Cont-GRU (DP)."

As shown in Table 6, perturbing the hidden state brings more significant influences to the tested downstream tasks, which means that our DDE-based formulation plays a crucial role in those tasks in comparison with the interpolated continuous path $\mathbf{x}(t)$.

## 5   CONCLUSIONS AND LIMITATIONS

We present the first fully continuous GRU model. The hidden state $\mathbf{h}(t)$ had been continuously generalized by existing methods. However, to our knowledge, Cont-GRU is the first model to successively generalize all the parts (gates) of GRUs, including the hidden state. To this end, we rely on interpolation methods to reconstruct a continuous path from a discrete time series sample. We then define a DDE-based model to interpret GRUs in a continuous manner. In our experiment with 5 real-world datasets and 17 baselines, our method consistently shows the best accuracy. Interestingly, other piece-wise continuous models generalizing the hidden state only do not work well in some cases where our fully continuous model works well. We consider that these experimental results well prove the efficacy of the fully continuous model.

**Limitations**   Our model shows good performance, but there exists room for improvement. For example, in the USHCN dataset, our model performs well at forecasting sudden changes, but its absolute error scale is not always satisfactory. For some test cases, in addition, all GRU-based models, including GRU-ODE-Bayes and our Cont-GRU, are not successful. We think that GRUs are not suitable for processing the test cases. However, it is hard to say that this is a limitation for our model since it is common for all GRU-based models.

**Reproducibility Statement** To ensure the reproducibility and completeness of this paper, we make our code available at https://drive.google.com/drive/folders/1pPKNlQFznxnFmE0-1ar3o6-k0bQ2sjjd?usp=sharing. We give details on our experimental protocol in the Appendix C.

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
