# A  WHY SHOULD GRUS BE FORMULATED AS DDES?

There are several attempts to interpret GRUs in a continuous manner for solving problems such as time series forecasting, classification, and so forth (Mozer et al., 2017; Jordan et al., 2021). This shows how meaningful those attempts are in the field of time series. All these continuous interpretations of GRUs are robust to irregular time series processing to some degree although not perfect. Therefore, one can use a trained continuous model for real-world environments where time series observations can be missing from time to time (due to malfunctioning sensors and/or communication channels while collecting data).

In addition, our Cont-GRU, the first DDE-based continuous modeling of GRUs, has a couple of strong points in comparison with existing methods: i) Our DDE-based modeling does not have the homeomorphic limitation of ODEs. In all the tasks in our experiments, our method significantly outperforms existing methods. ii) Our method has smaller empirical memory usage and training time than those of existing continuous models (as reported in our experimental results). Owing to the increased model capability by DDEs, our model size is relatively smaller than others, resulting in lightweight empirical processing (cf. Appendix M).

In the design philosophy of GRU cells, previous information is used to determine current output, i.e., long/short-term dependencies, and this concept is closely related to that of DDEs. Since DDEs are to model time-delay systems where long/shot-term past information influences current output, we believe that GRUs should be continuously interpreted in DDEs and our experimental results justify it.

# B  BASELINES

We test the following state-of-the-art baselines to compare our proposed Cont-GRU with:

1. RNN, LSTM (Hochreiter & Schmidhuber, 1997), and GRU (Chung et al., 2014) are all recurrent neural network-based models that can process sequential data. LSTM is designed to learn long-term dependencies and GRU uses gating mechanisms to control the flow of information.

2. NODE (Chen et al., 2018) is a continuous-time model that defines the hidden state $\mathbf{h}(t)$ with an initial value problem (IVP).

3. ODE-RNN is a combination of RNN and NODE. When a NODE evolves a hidden state $\mathbf{h}(t)$ continuously between observations, an RNN Cell causes that $\mathbf{h}(t)$ jumps to another hidden state when a new observation arrives.

4. GRU-$\Delta t$ and GRU-D are advanced versions of GRU to process irregular time series. GRU-$\Delta t$ is a GRU that additionally takes the time difference between observations as an input. GRU-D (Che et al., 2018) is a modified version of GRU-$\Delta t$ with learnable exponential decay between observations.

5. GRU-ODE is similar to NODE. The only difference is that it derived an ODE function from GRU. GRU-ODE is piece-wise continuous through all time.

6. Latent-ODE is a model that can explain the latent state with ODEs, and is a suitable model for time series prediction.

7. Augmented-ODE is the method proposed by (Dupont et al., 2019), which inserts zeros to the ODE state of Latent-ODE.

8. ACE-NODE (Jhin et al., 2021a) is an attention-based Neural ODE model.

9. GRU-ODE-Bayes (Brouwer et al., 2019) is a combination of GRU-ODE and GRU-Bayes. GRU-ODE calculates a hidden state $\mathbf{h}(t)$ in a continuous manner between observations, and GRU-Bayes is used to discretely jump $\mathbf{h}(t)$ to another state when a new observation arrives. GRU-ODE-Bayes is often used to predict sporadically observed data. However, those models that combine NODEs with jumps, such as ODE-RNN and GRU-ODE-Bayes, are not completely continuous but piece-wise continuous.

10. Neural jump ODE (NJODE) (Herrera et al., 2021), on the other hand, is a data-driven approach that continuously learns the conditional expectations of a probabilistic process.

11. Neural CDE (NCDE (Kidger et al., 2020)) is a conceptually enhanced model of NODEs based on the theory of controlled differential equations.

12. Attentive Neural CDE (ANCDE) (Jhin et al., 2021b) is an attention-based NCDE model.

13. Extrapolation and interpolation-based Neural CDE (EXIT) (Jhin et al., 2022) is based on both the interpolation and the extrapolation of time series when making the continuous data path.

14. SCINet (Liu et al., 2021) is a novel neural network model that performs sample convolution and interaction for temporal modeling and shows excellent performance in time series tasks.

The best hyperparameters of our model and baselines are reported in Appendix C. Some official codes are the following links:

1. NODE : `https://github.com/rtqichen/torchdiffeq`,

2. ODE-RNN, GRU-$\Delta t$, GRU-D, GRU-ODE and NCDE : `https://github.com/patrick-kidger/NeuralCDE`,

3. Latent-ODE, Augmented-ODE : `https://github.com/YuliaRubanova/latent_ode`,

4. ACE-NODE : `https://github.com/sheoyon-jhin/ACE-NODE`,

5. GRU-ODE-Bayes : `https://github.com/edebrouwer/gru_ode_bayes`,

6. NJODE : `https://github.com/HerreraKrachTeichmann/NJODE`,

7. ANCDE : `https://github.com/sheoyon-jhin/ANCDE`,

8. EXIT : `https://github.com/sheoyon-jhin/EXIT`,

9. SCINet : `https://github.com/cure-lab/SCINet`.

## C  HYPERPARAMETERS

For the best outcome of baselines and our method, we conduct hyperparameter search for them based on the recommended hyperparameter set from each paper. Considered hyperparameter sets are as follows:

### C.1  USHCN

In USHCN, we train for 150 epochs with a batch size of 256, and stop early if the training loss doesn't decrease for 50 epochs. A hidden size in $\{19, 29, 39, 49, 59, 69\}$ and a learning rate $\lambda$ in $\{1.0 \times e^{-3}, 5.0 \times e^{-3}, 1.0 \times e^{-2}, 5.0 \times e^{-2}\}$ are used. We used DOPRI5 as an ODE solver.

### C.2  PHYSIONET SEPSIS

In PhysioNet Sepsis, we train for 100 epochs with a batch size of 1,024, and stop early if the training loss doesn't decrease for 50 epochs. A hidden vector size in $\{29, 39, 49, 59, 69\}$ and a learning rate $\lambda$ in $\{1.0 \times e^{-3}, 5.0 \times e^{-3}, 1.0 \times e^{-2}, 5.0 \times e^{-2}\}$ are used. We used DOPRI5 as an ODE solver.

### C.3  GOOGLE STOCK

In Google Stock, we train for 200 epochs with a batch size of 256, and stop early if the training loss doesn't decrease for 100 epochs. We use a hidden vector size of $\{15, 25, 35, 45\}$, and a learning rate $\lambda$ of $\{1.0 \times e^{-4}, 5.0 \times e^{-4}, 1.0 \times e^{-3}, 5.0 \times e^{-3}\}$. We used DOPRI5 as an ODE solver.

### C.4  ETT DATASETS

In ETT, we train for 300 epochs with a batch size of 256, and stop early if the training loss doesn't decrease for 100 epochs. We use a hidden vector size of $\{19, 29, 39, 49\}$, and a learning rate $\lambda$ of $\{1.0 \times e^{-4}, 5.0 \times e^{-4}, 1.0 \times e^{-3}, 5.0 \times e^{-3}\}$. We used RK4 as an ODE solver.

# D BEST HYPERPARAMETERS

For reproducibility, we report the best hyperparameters as follows:

## D.1 USHCN

1. RNN-based models: For RNN, LSTM, and GRU, we use $\lambda = 5.0 \times e^{-3}$, hidden size = 49, and weight decay = $1.0 \times e^{-3}$.

2. GRU-based models: For GRU-$\Delta t$, GRU-D, we use $\lambda = 5.0 \times e^{-3}$, hidden size = 49, and weight decay = $1.0 \times e^{-3}$.

3. ODE-based models: For NODE, ODE-RNN, and GRU-ODE, we use $\lambda = 5.0 \times e^{-3}$, hidden size = 49, and weight decay = $1.0 \times e^{-3}$. For Latent-ODE, Augmented-ODE, ACE-NODE, we use $\lambda = 1.0 \times e^{-3}$, hidden size = 69, and weight decay = $1.0 \times e^{-4}$. For GRU-ODE-Bayes and NJODE, we use $\lambda = 5.0 \times e^{-3}$, hidden size = 69, and weight decay = $1.0 \times e^{-3}$.

4. CDE-based modes: For NCDE, ANCDE, and EXIT, we use $\lambda = 5.0 \times e^{-3}$, hidden size = 19, and weight decay = $1.0 \times e^{-4}$.

5. For SCINet, we use $\lambda = 1.0 \times e^{-4}$, hidden size of convolution = 7, and weight decay = $1.0 \times e^{-4}$.

6. For Cont-GRU, we use $\lambda = 5.0 \times e^{-3}$, hidden size = 19, and weight decay = $1.0 \times e^{-4}$.

## D.2 PHYSIONET SEPSIS

1. GRU-based models: For GRU-$\Delta t$, GRU-D, we use $\lambda = 1.0 \times e^{-2}$, hidden size = 49, and weight decay = $1.0 \times e^{-3}$.

2. ODE-based models: For NODE, ODE-RNN, and GRU-ODE, we use $\lambda = 1.0 \times e^{-3}$, hidden size = 49, and weight decay = $1.0 \times e^{-3}$. For Latent-ODE, Augmented-ODE, ACE-NODE, we use $\lambda = 1.0 \times e^{-3}$, hidden size = 69, and weight decay = $1.0 \times e^{-4}$. For GRU-ODE-Bayes and NJODE, we use $\lambda = 5.0 \times e^{-3}$, hidden size = 69, and weight decay = $1.0 \times e^{-3}$.

3. CDE-based modes: For NCDE, ANCDE, and EXIT, we use $\lambda = 5.0 \times e^{-3}$, hidden size = 19, and weight decay = $1.0 \times e^{-4}$.

4. For Cont-GRU, we use $\lambda = 1.0e^{-3}$, hidden size = 59, and weight decay = $1.0e^{-2}$.

## D.3 GOOGLE STOCK

1. ODE-based models: For NODE, ODE-RNN, and GRU-ODE, we use $\lambda = 1.0e^{-2}$, hidden size = 25, and weight decay = $1.0e^{-4}$. For Latent-ODE, Augmented-ODE, ACE-NODE, we use $\lambda = 1.0 \times e^{-4}$, hidden size = 45, and weight decay = $1.0 \times e^{-4}$. For GRU-ODE-Bayes and NJODE, we use $\lambda = 5.0 \times e^{-4}$, hidden size = 45, and weight decay = $1.0 \times e^{-3}$.

2. CDE-based models: For NCDE, ANCDE, and EXIT, we use $\lambda = 1.0 \times e^{-4}$, hidden size = 35, and weight decay = $1.0 \times e^{-4}$.

3. For SCINet, we use $\lambda = 1.0 \times e^{-4}$, hidden size of convolution = 7, and weight decay = $1.0 \times e^{-4}$.

4. For Cont-GRU, we use $\lambda = 1.0 \times e^{-2}$, hidden size = 25, and weight decay = $1.0 \times e^{-4}$.

## D.4 ETT

1. RNN-based models: For RNN, LSTM, and GRU, we use $\lambda = 1.0 \times e^{-3}$, hidden size = 29, and weight decay = $1.0 \times e^{-4}$.

2. GRU-based models: For GRU-$\Delta t$, GRU-D, we use $\lambda = 1.0 \times e^{-3}$, hidden size = 49, and weight decay = $1.0 \times e^{-4}$.

3. ODE-based models: For NODE, ODE-RNN, and GRU-ODE, we use $\lambda = 5.0 \times e^{-3}$, hidden size = 49, and weight decay = $1.0e^{-3}$. For Latent-ODE, Augmented-ODE, ACE-NODE, we use $\lambda = 1.0 \times e^{-3}$, hidden size = 59, and weight decay = $1.0 \times e^{-4}$. For

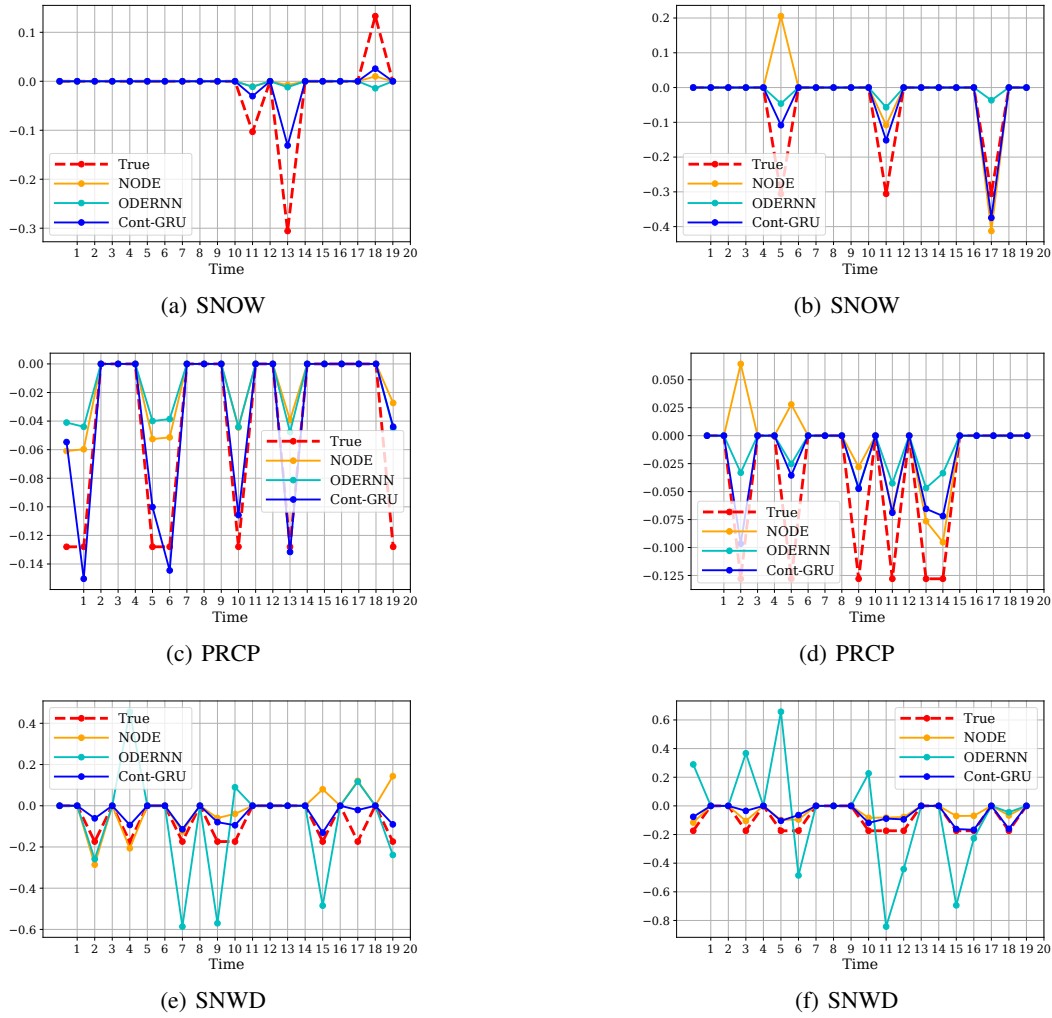

Figure 3: Forecasting visualization on USHCN

GRU-ODE-Bayes and NJODE, we use $\lambda = 5.0 \times e^{-3}$, hidden size = 59, and weight decay $= 1.0 \times e^{-3}$.

4. CDE-based models: For NCDE, ANCDE, and EXIT, we use $\lambda = 1.0 \times e^{-4}$, hidden size = 49, and weight decay $= 1.0 \times e^{-4}$.

5. For SCINet, we use $\lambda = 1.0 \times e^{-4}$, hidden size of convolution = 7, and weight decay $= 1.0 \times e^{-4}$.

6. For Cont-GRU, we use $\lambda = 1.0 \times e^{-3}$, hidden size = 49, and weight decay $= 1.0 \times e^{-4}$.

# E   VISUALIZATION ON USHCN

Figures 3 and 4 visualize the prediction results for snowfall (SNOW), precipitation (PRCP), snow depth (SNWD), maximum temperature (TMAX), and minimum temperature (TMIN). It can be seen that our model in blue line predicts trends better than the top-2 baseline models, ODE-RNN and NODE.

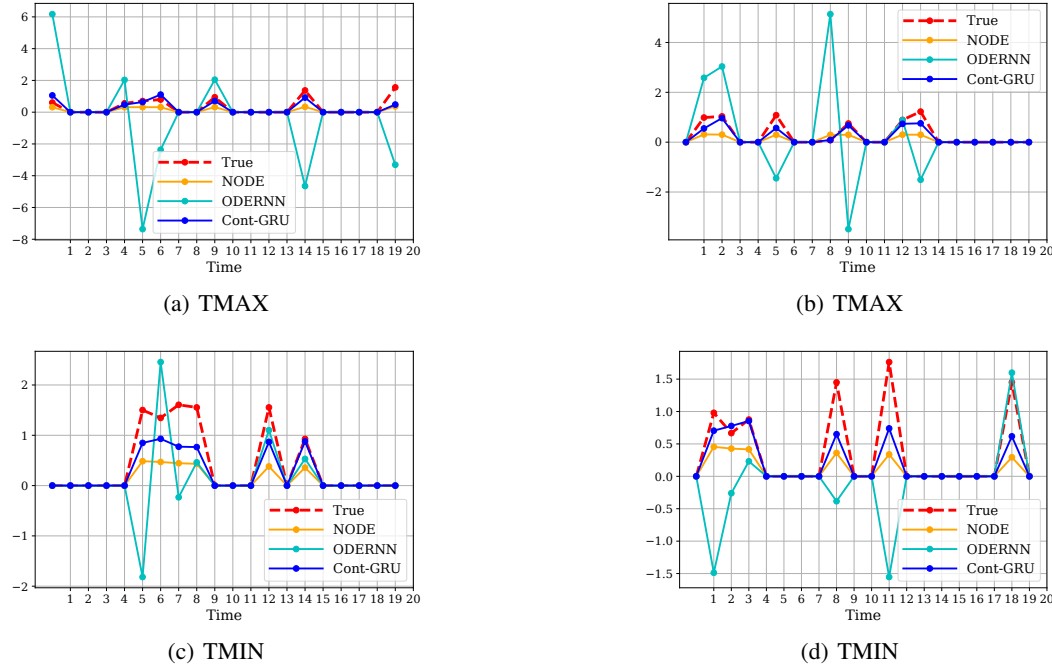

Figure 4: Forecasting visualization on USHCN

## F   VISUALIZATION ON GOOGLE STOCK

We visualize the forecasting results in Google Stock for various periods and various variables in Figures 5 and 6.

## G   VALUES IN THE RESET GATE

In Figure 7, we visualize the results of the role of the reset gate. We visualize for the 20-day period i) from November 13, 2017 to December 11, 2017, and ii) from May 7, 2021 to June, 4 2021 in Google Stock.

## H   DETAILED DESCRIPTIONS OF INTERPOLATION METHODS

A cubic spline is a spline of piece-wise cubic polynomials passing through $m$ given control points. A second order derivative of zero at the endpoints is assumed to solve for a cubic spline. This provides boundary conditions that allow the two equations to be solved with respect to the $m - 2$ equation. The system of equations for a cubic spline in one dimension can be written as:

$$S_i(x) = a_i + b_i * x + c_i * x^2 + d_i x^3. \tag{8}$$

For the function $y = \mathbf{f}(x)$, we take a set of points $[x_i, y_i]$ where $i \in \{0, 1, 2, ..., n\}$. It is a piece-wise continuous curve that passes through each value in the table for cubic spline interpolation. The prerequisites for the spline of degree $K = 3$ are as follows: first, the domain of $s$ must lie inside the intervals of $[a, b]$; second, all three parameters must be continuous functions on $[a, b]$.

$$S(x) = \{S_i(x) | x \in [t_{i-1}, t_i], i \in [1, n)\} \tag{9}$$

$S_i(x)$ is defined as a cubic polynomial in the sub interval $[x_i, x_{i+1}]$. We need $4n$ parameters to solve the spline because there are $n$ intervals and 4 coefficients in each equation. We may derive the $2n$ equation from the requirement that each cubic spline equation meet the value at both ends:

$$S_i(x_i) = y_i, \tag{10}$$
$$S_i(x_{i+1}) = y_{i+1}. \tag{11}$$

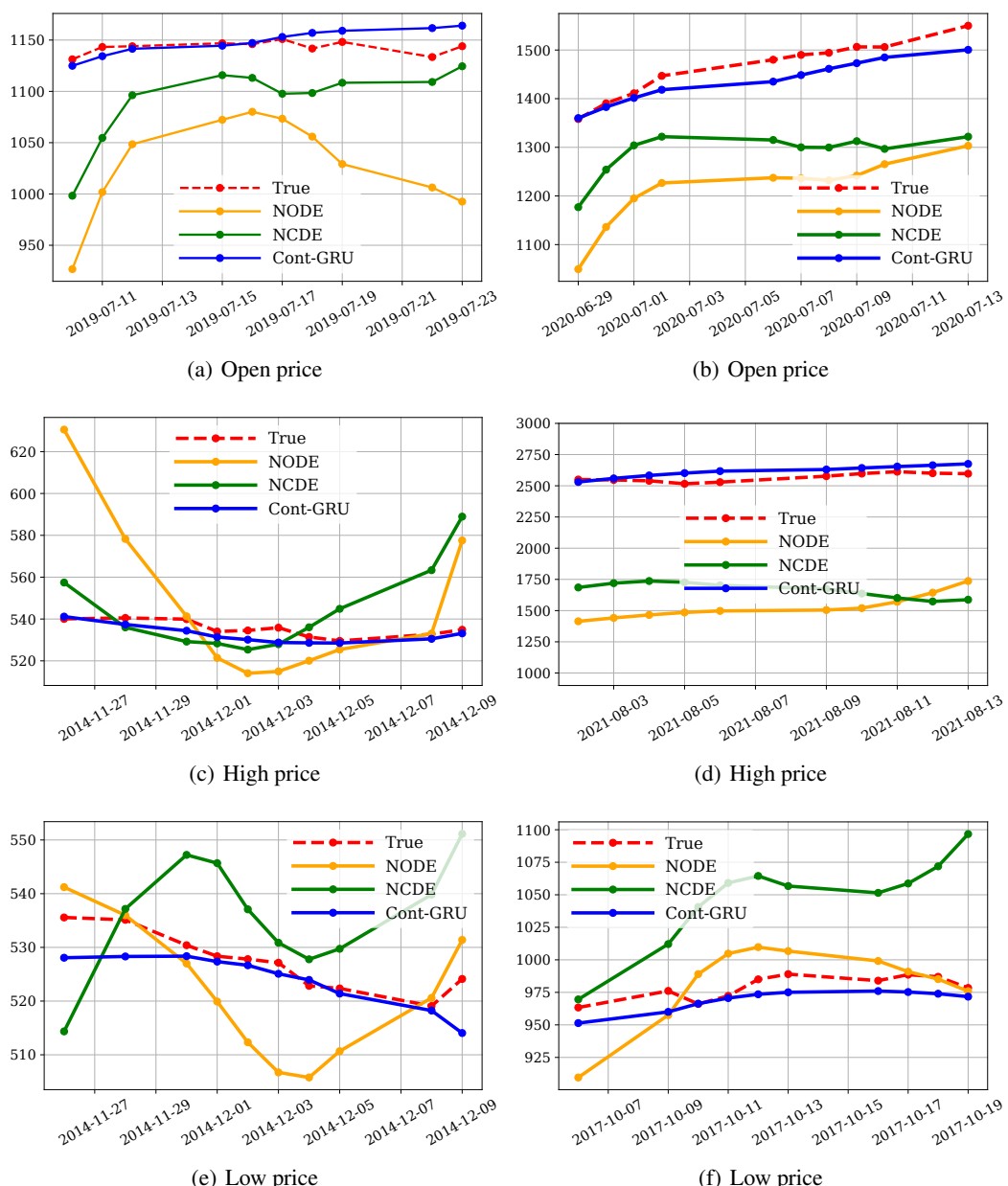

Figure 5: Forecasting visualization on Google Stock

In addition to being continuous and differentiable, the aforementioned cubic spline equations should also have defined first and second derivatives that are continuous on control points:

$$S'_{i-1}(x_i) = S'_i(x_i), \tag{12}$$

$$S'_{i-1}(x_i) = S'_i(x_i). \tag{13}$$

The $2n - 2$ equation restrictions are given for $\{1, 2, 3, ..., n - 1\}$. Thus, two additional equations are required to solve the above cubic spline. We'll employ some natural boundary conditions for it. We assume that the second derivative of the spline at boundary points is zero in the Natural Cubic

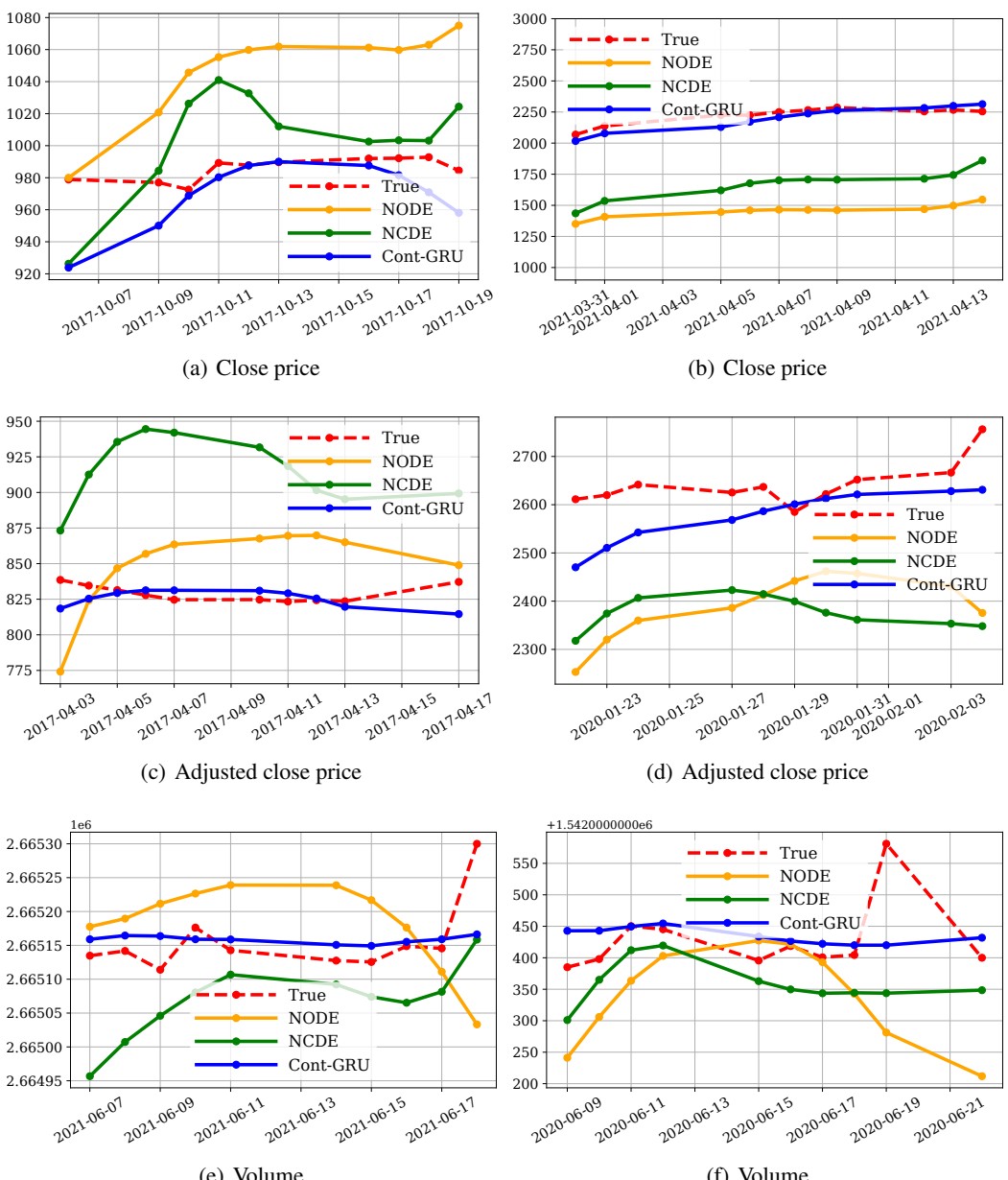

Figure 6: Forecasting visualization on Google Stock

Spline:

$$S''(x_0) = 0, \tag{14}$$

$$S''(x_n) = 0. \tag{15}$$

Now that we know that $S(x)$ is a third-order polynomial, we may infer that $S''(x)$ is an interpolating linear spline. As a result, we first create $S''(x)$ before twice integrating it to get $S(x)$. Now, let's assume that $z_i = S''(x_i), i \in \{0, 1, 2..n\}$ and from the natural boundary condition $z_0 = z_n = 0$ that $t_i = x_i$ for $i \in \{0, 1, ...n\}$. A linear spline is produced by twice differentiating a cubic spline,

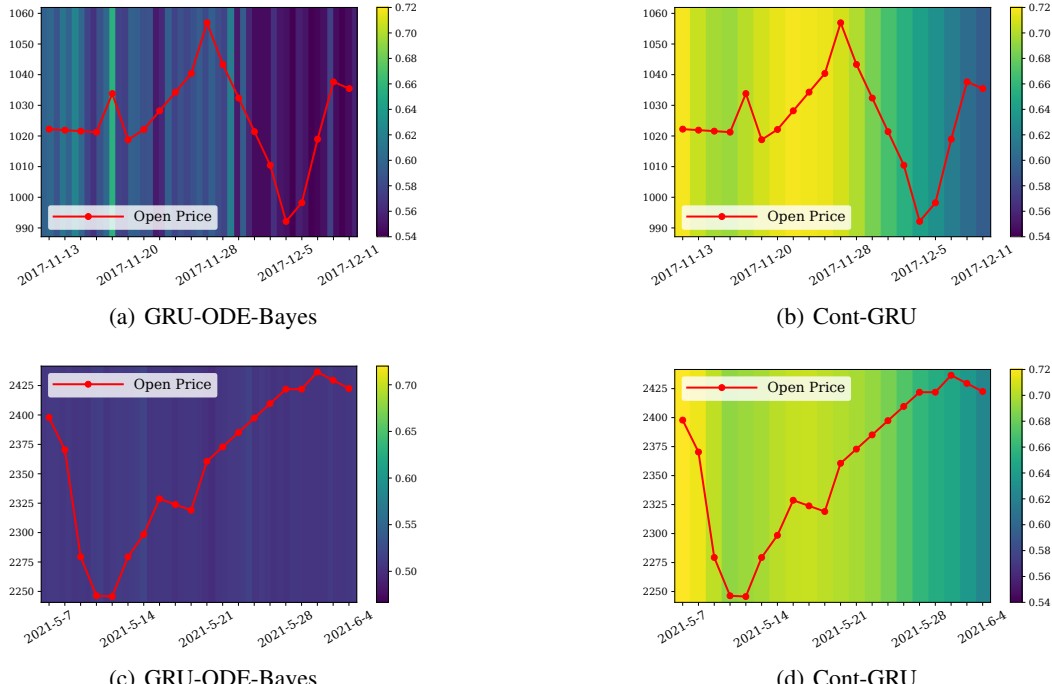

(a) GRU-ODE-Bayes

(b) Cont-GRU

(c) GRU-ODE-Bayes

(d) Cont-GRU

Figure 7: Values in the reset gate

and it has the following notation:

$$S_i''(x) = z_i \frac{x - t_{i+1}}{t_i - t_{i+1}} + z_{i+1} \frac{x - t_i}{t_{i+1} - t_i}, \tag{16}$$

where $\Delta t_i = t_{i+1} - t_i; t \in \{0, 1, 2, ..., n\}$. The equation is written as : $S''(x) = z_{i+1} \frac{x - t_i}{\Delta t_i} + z_i \frac{t_{i+1} - x}{\Delta t_i}$. Integrating this equation twice we get a cubic spline.

$$S(x) = \frac{z_{i+1}}{6\Delta t_i}(x - t_i)^3 + \frac{z_i}{6\Delta t_i}(t_{i+1} - x)^3 + \\ C_i(x_i - t) + D_i(t_{i+1} - x), \tag{17}$$

where

$$C_i = \frac{y_{i+1}}{\Delta t_i} - \frac{z_{i+1} * \Delta t_i}{6}, \tag{18}$$

$$D_i = \frac{y_i}{\Delta t_i} - \frac{z_i * \Delta t_i}{6}. \tag{19}$$

Now, verify the derivative at $t_i$ is continuous. We must first locate the derivative and add the following condition:

$$S'(x) = \frac{z_{i+1}}{2\Delta t_i}(x - t_i)^2 - \frac{z_i}{2\Delta t_i}(t_{i+1} - x)^2 \\ + \frac{1}{\Delta t_i}(y_{i+1} - y_i) - \frac{\Delta t_i}{6}(z_{i+1} - z_i), \tag{20}$$

where $b_i = \frac{1}{\Delta t_i}(y_{i+1} - y_i)$. The following equation results from calculating the continuity equation mentioned above:

$$6(b_i - b_{i-1}) = \Delta t_{i-1} z_{i-1} + 2(\Delta t_{i-1} + \Delta t_i)z_i + \Delta t_i z_{i+1}. \tag{21}$$

Since there are $n$ parameters of our $z$, and $n$ systems of equations we know, we can find $z$ by solving the system of equations.

# I  DERIVATIVES OF $\mathbf{z}(t), \mathbf{g}(t), \mathbf{r}(t)$

**Time derivative of $\mathbf{z}(t)$**  The *continuous* update gate is written as $\mathbf{z}(t) = \sigma\big(\mathbf{W}_z\mathbf{x}(t) + \mathbf{U}_z\mathbf{h}(t - \tau) + \mathbf{b}_z\big) = \sigma(\mathbf{A}(t, t - \tau))$, and its derivative, denoted $\frac{d\mathbf{z}(t)}{dt}$, is as follows:

$$\frac{d\mathbf{z}(t)}{dt} = \sigma\big(\mathbf{A}(t, t - \tau)\big)(1 - \sigma(\mathbf{A}(t, t - \tau)))\frac{d\mathbf{A}(t, t - \tau)}{dt}, \tag{22}$$

where $\mathbf{A}(t, t - \tau) = \mathbf{W}_z\mathbf{x}(t) + \mathbf{U}_z\mathbf{h}(t - \tau) + \mathbf{b}_z$, and $\frac{d\mathbf{A}(t, t-\tau)}{dt} = \mathbf{W}_z\frac{d\mathbf{x}(t)}{dt} + \mathbf{U}_z\frac{d\mathbf{h}(t-\tau)}{dt}$.

**Time derivative of $\mathbf{g}(t)$**  The *continuous* update vector has the form of $\mathbf{g}(t) = \phi\big(\mathbf{W}_g\mathbf{x}(t) + \mathbf{U}_g\big(\mathbf{r}(t) \odot \mathbf{h}(t - \tau)\big) + \mathbf{b}_g\big) = \phi(\mathbf{B}(t, t - \tau)$, and its derivative, $\frac{d\mathbf{g}(t)}{dt}$, can be calculate as follows:

$$\frac{d\mathbf{g}(t)}{dt} = \big(1 - \phi^2(\mathbf{B}(t, t - \tau)\big)\frac{d\mathbf{B}(t, t - \tau)}{dt}, \tag{23}$$

where $\mathbf{B}(t, t-\tau) = \mathbf{W}_g\mathbf{x}(t) + \mathbf{U}_g\big(\mathbf{r}(t)\odot\mathbf{h}(t-\tau)\big) + \mathbf{b}_g$, and $\frac{d\mathbf{B}(t,t-\tau)}{dt} = \mathbf{W}_g\frac{d\mathbf{x}(t)}{dt} + \mathbf{U}_g\frac{d\mathbf{r}(t)}{dt}\mathbf{h}(t-\tau) + \mathbf{U}_g\mathbf{r}(t)\frac{d\mathbf{h}(t-\tau)}{dt}$.

**Time derivative of $\mathbf{r}(t)$**  The *continuous* reset gate is defined as $\mathbf{r}(t) = \sigma\big(\mathbf{W}_r\mathbf{x}(t) + \mathbf{U}_r\mathbf{h}(t - \tau) + \mathbf{b}_r\big)$, and its derivative $\frac{d\mathbf{r}(t)}{dt}$ is derived as follows:

$$\frac{d\mathbf{r}(t)}{dt} = \sigma\big(\mathbf{C}(t)\big)(1 - \sigma(\mathbf{C}(t, t - \tau)))\frac{d\mathbf{C}(t, t - \tau)}{dt}, \tag{24}$$

where $\mathbf{C}(t, t - \tau) = \mathbf{W}_r\mathbf{x}(t) + \mathbf{U}_r\mathbf{h}(t - \tau) + \mathbf{b}_r$, and $\frac{d\mathbf{C}(t,t-\tau)}{dt} = \mathbf{W}_r\frac{d\mathbf{x}(t)}{dt} + \mathbf{U}_r\frac{d\mathbf{h}(t-\tau)}{dt}$.

# J  PROOF OF EQUATION  EQUATION 6

First, let $\mathbf{z}(t)$, $\mathbf{g}(t)$, and $\mathbf{r}(t)$ be the update gate, the update vector, and the reset gate of GRUs:

$$\begin{aligned}
\mathbf{z}(t) &= \sigma\big(\mathbf{W}_z\mathbf{x}(t) + \mathbf{U}_z\mathbf{h}(t - \tau) + \mathbf{b}_z\big), \\
\mathbf{g}(t) &= \phi\big(\mathbf{W}_g\mathbf{x}(t) + \mathbf{U}_g\big(\mathbf{r}(t) \odot \mathbf{h}(t - \tau)\big) + \mathbf{b}_g\big), \\
\mathbf{r}(t) &= \sigma\big(\mathbf{W}_r\mathbf{x}(t) + \mathbf{U}_r\mathbf{h}(t - \tau) + \mathbf{b}_r\big),
\end{aligned} \tag{25}$$

To simplify the equations, we will define them as follows:

$$\begin{aligned}
\mathbf{z}(t) &= \sigma\big(\mathbf{A}(t, t - \tau)\big), \\
\mathbf{g}(t) &= \phi\big(\mathbf{B}(t, t - \tau)\big), \\
\mathbf{r}(t) &= \sigma\big(\mathbf{C}(t, t - \tau)\big),
\end{aligned} \tag{26}$$

where $\mathbf{A}(t, t-\tau) = \mathbf{W}_z\mathbf{x}(t) + \mathbf{U}_z\mathbf{h}(t-\tau) + \mathbf{b}_z$, $\mathbf{B}(t, t-\tau) = \mathbf{W}_h\mathbf{x}(t) + \mathbf{U}_h\big(\mathbf{r}(t)\odot\mathbf{h}(t-\tau)\big) + \mathbf{b}_h$, and $\mathbf{C}(t, t - \tau) = \mathbf{W}_r\mathbf{x}(t) + \mathbf{U}_r\mathbf{h}(t - \tau) + \mathbf{b}_r$. The derivatives of $\mathbf{z}(t)$, $\mathbf{g}(t)$, and $\mathbf{r}(t)$ are defined as follows:

$$\begin{aligned}
\frac{d\mathbf{z}(t)}{dt} &= \sigma(\mathbf{A}(t, t - \tau))(1 - \sigma(\mathbf{A}(t, t - \tau)))\frac{d\mathbf{A}(t, t - \tau)}{dt} \\
\frac{d\mathbf{g}(t)}{dt} &= (1 - \phi^2(\mathbf{B}(t, t - \tau)))\frac{d\mathbf{B}(t, t - \tau)}{dt} \\
\frac{d\mathbf{r}(t)}{dt} &= \sigma(\mathbf{C}(t, t - \tau))(1 - \sigma(\mathbf{C}(t, t - \tau)))\frac{d\mathbf{C}(t, t - \tau)}{dt}
\end{aligned} \tag{27}$$

Lastly, the hidden state $\mathbf{h}(t)$ of GRUs is written as follows:

$$\mathbf{h}(t) = \mathbf{z}(t) \odot \mathbf{h}(t - \tau) + (1 - \mathbf{z}(t)) \odot \mathbf{g}(t). \tag{28}$$

The derivative of the hidden state $\mathbf{h}(t)$ is defined by the chain rule as follows:

$$
\begin{aligned}
\frac{d\mathbf{h}(t)}{dt} &= \frac{d\mathbf{z}(t)}{dt} \odot \mathbf{h}(t-\tau) + \mathbf{z}(t) \odot \frac{d\mathbf{h}(t-\tau)}{dt} \\
&\quad - \frac{d\mathbf{z}(t)}{dt} \odot \mathbf{g}(t) + (1 - \mathbf{z}(t)) \odot \frac{d\mathbf{g}(t)}{dt}, \\
&= \frac{d\mathbf{z}(t)}{dt} \odot \big(\mathbf{h}(t-\tau) - \mathbf{g}(t)\big) \\
&\quad + \mathbf{z}(t) \odot \big(\frac{d\mathbf{h}(t-\tau)}{dt} - \frac{d\mathbf{g}(t)}{dt}\big) + \frac{d\mathbf{g}(t)}{dt}, \\
&= \frac{d\mathbf{z}(t)}{dt} \odot \zeta(t, t-\tau) + \mathbf{z}(t) \odot \frac{d\zeta(t, t-\tau)}{dt} + \frac{d\mathbf{g}(t)}{dt},
\end{aligned}
\tag{29}
$$

where $\zeta(t, t-\tau) = \mathbf{h}(t-\tau) - \mathbf{g}(t)$. So, we can rewrite $\frac{d\mathbf{h}(t)}{dt}$ as follows:

$$
\frac{d\mathbf{h}(t)}{dt} = \frac{d(\mathbf{z}(t) \odot \zeta(t, t-\tau))}{dt} + \frac{d\mathbf{g}(t)}{dt}
\tag{30}
$$

## K  DATA PREPROCESSING DETAILS

The datasets used in our experiments are publicly available and can be downloaded at the following locations:

1. USHCN: https://cdiac.ess-dive.lbl.gov/ftp/ushcn_daily/,
2. PhysioNet Sepsis[2]: https://physionet.org/content/challenge-2019/1.0.0/,
3. Google Stock: https://finance.yahoo.com/quote/GOOG/history?p=GOOG,
4. ETT: https://github.com/zhouhaoyi/ETDataset.

We split the entire dataset into training/validating/testing parts. The first 70% of the data is used as training, 15% is used for validating, and the last 15% is used for testing.

### K.1  USHCN

We follow the preprocessing process of GRU-ODE-Bayes (Brouwer et al., 2019). We look at 128 time sequences for training and forecast next 16,32,64 time sequences. And the stride interval is 64 time sequences.

### K.2  PHYSIONET SEPSIS

We follow the preprocessing settings of NCDE (Kidger et al., 2020). NCDE used a new variable, called observation intensity (OI), for learning. The observation intensity (OI) is the frequency of observations.

### K.3  GOOGLE STOCK

We used the Google Stock data from 2011 to 2021. Since the scale of each variable is different, normalization was performed between 0 and 1. We look at 20 time sequences for training and forecast next 10 time sequences. And the stride interval is 5 time sequences.

### K.4  ETT

We follow the preprocessing process of SCINet (Liu et al., 2021).

---

[2]This dataset follows the license policy of CC-BY 4.0.

## L    COMPUTING INFRASTRUCTURES

In this section, we describe our software/hardware environments. All experiments were conducted in the following software and hardware environments: UBUNTU 18.04 LTS, PYTHON 3.8, NUMPY 1.21.5, SCIPY 1.7.3, MATPLOTLIB 3.3.1, PYTORCH 1.7.1, CUDA 11.0, GEFORCE RTX 3090. We repeat the training and testing procedures with five different random seeds and report their mean and standard deviation accuracy.

## M    MEMORY USAGE AND TIME COST

In Tables 7 and 8, we report the memory usage and training time per epoch of all baselines and our proposed method.

Table 7: Memory usage and training time on USHCN,Google Stock, and PhysioNet Sepsis

| Models | USHCN | | Google Stock | | PhysioNet Sepsis | |
|---|---|---|---|---|---|---|
| | Memory Usage (MB) | Training Time (s) | Memory Usage (MB) | Training Time (s) | Memory Usage (MB) | Training Time (s) |
| GRU | 34.12 | 4.01 | — | — | — | — |
| LSTM | 36.53 | 3.77 | — | — | — | — |
| RNN | 38.31 | 5.44 | — | — | — | — |
| NODE | 55.79 | 30.6 | 4.23 | 1.14 | 177.8 | 51.8 |
| ODE-RNN | 565.1 | 9.14 | 5.95 | 0.35 | 387.3 | 10.8 |
| GRU-$\Delta t$ | 492.7 | 1.26 | 2.35 | 0.02 | 352.5 | 2.03 |
| GRU-D | 528.9 | 1.50 | 3.33 | 0.02 | 369.8 | 2.45 |
| GRU-ODE | 76.79 | 8.63 | 7.93 | 0.28 | 178.8 | 15.5 |
| Latent-ODE | 210.3 | 15.2 | 17.3 | 4.17 | 207.8 | 314 |
| Augmented-ODE | 197.2 | 46.5 | 21.3 | 4.26 | 742.3 | 322 |
| ACE-NODE | 210.4 | 44.7 | 22.6 | 4.88 | 184.5 | 212 |
| GRU-ODE-Bayes | 1,613 | 33.7 | 24.2 | 4.83 | 456.3 | 104 |
| NJODE | 1,889 | 36.1 | 20.8 | 3.48 | 401.4 | 113 |
| NCDE | 62.32 | 251 | 9.36 | 1.37 | 204.4 | 191 |
| ANCDE | 121.7 | 267 | 10.3 | 3.42 | 247.7 | 181 |
| EXIT | 182.3 | 271 | 14.7 | 4.01 | 257.2 | 211 |
| SCINet | 144.6 | 127 | 16.7 | 6.11 | – | – |
| Cont-GRU | 67.85 | 20.2 | 8.89 | 0.79 | 177.9 | 49.2 |

Table 8: Memory usage and training time on ETTh1, and ETTh2

| Models | ETTh1 | | ETTh2 | |
|---|---|---|---|---|
| | Memory Usage (MB) | Training Time (s) | Memory Usage (MB) | Training Time (s) |
| GRU | 4,573 | 0.006 | 4,573 | 0.007 |
| LSTM | 4,573 | 0.004 | 4,573 | 0.004 |
| RNN | 3,102 | 0.007 | 3,102 | 0.008 |
| NODE | 69.60 | 41.00 | 69.60 | 52.04 |
| ODE-RNN | 361.4 | 15.20 | 361.4 | 23.78 |
| GRU-$\Delta t$ | 316.1 | 5.037 | 316.1 | 4.077 |
| GRU-D | 338.6 | 5.389 | 338.6 | 4.771 |
| GRU-ODE | 167.5 | 19.48 | 167.5 | 22.44 |
| Latent-ODE | 210.1 | 314.2 | 201.1 | 322.7 |
| Augmented-ODE | 504.2 | 250.7 | 504.2 | 242.9 |
| ACE-NODE | 287.1 | 240.1 | 287.1 | 256.1 |
| GRU-ODE-Bayes | 302.1 | 115.6 | 302.1 | 134.7 |
| NJODE | 267.2 | 237.1 | 267.2 | 212.4 |
| NCDE | 81.10 | 268.8 | 81.10 | 243.4 |
| ANCDE | 124.1 | 50.12 | 124.1 | 60.10 |
| EXIT | 159.1 | 60.12 | 159.1 | 61.88 |
| SCINet | 16.07 | 13.33 | 16.07 | 23.12 |
| Cont-GRU | 178.8 | 28.42 | 178.8 | 62.29 |