# OpenReview forum: "Cont-GRU: Fully Continuous Gated Recurrent Units for Irregular Time Series"
_ICLR.cc/2024/Conference — ICLR 2024 Conference Withdrawn Submission_

### Official Review · Reviewer_ZdKC · 2023-10-29

**Soundness:** 2 fair
**Presentation:** 2 fair
**Contribution:** 2 fair
**Rating:** 5
**Confidence:** 3

**Summary:**

This paper proposes and studies fully continuous GRUs by reformulating GRUs as delay differential equations. Empirical results are provided to justify the efficacy of the proposed model.

**Strengths:**

- Provide clean formulation and interpretations of a fully continuous version of GRUs
- Overall the paper is easy to follow

**Weaknesses:**

- Missing comparisons to related work that uses delay differential equations for designing RNNs; e.g., the $\tau$-GRU proposed in https://arxiv.org/abs/2212.00228, and also other recent models (e.g., LEM, coRNN, state-space models)
- Missing evaluation on important benchmark tasks such as sequential image classification on MNIST and CIFAR-10
- Missing ablation studies on the delay factor $\tau$ (what is the role of it?)
- Missing details on the activation functions written in Eq. (1) (what are $\sigma$ and $\phi$?)
- Missing theoretical results to show that the proposed model indeed has better representation learning capability

**Questions:**

- I understand that the DDE is solved using an ODE solver with adaptive step sizes but how does the behavior of the model depends on $\tau$ when it is fixed? What happen if $\tau$ is too big?
- What are the adaptive step sizes used for the experiments?

---

### Official Review · Reviewer_LsJ8 · 2023-10-30

**Soundness:** 3 good
**Presentation:** 3 good
**Contribution:** 3 good
**Rating:** 3
**Confidence:** 3

**Summary:**

The paper proposed a fully time-continuous version of the gated recurrent unit (GRU) RNN model. Different from the standard ODE-RNN of Rubanova et al. 2019, which is comprised of a continuous-time ODE and a discrete RNN, the Cont-GRU is fully continuous. The paper shows that the Cont-GRU outperforms alternative time-continuous architectures experimentally.

**Strengths:**

# Pros:
- Interesting and novel idea to make a discrete time RNN continuous while maintaining the same discrete time semantics by using time-delayed differential equations.
- The paper is mostly well written

**Weaknesses:**

# Cons:
- The experiments are primarily based on outdated RNN architectures and miss breakthrough time-continuous architectures of the past few years (Gu et al. 2021, Rusch et al. 2021, and Rusch et al. 2022).


# References
- Gu et al. 2021, Efficiently Modeling Long Sequences with Structured State Spaces.
- Rusch et al. 2021, Unicornn: A recurrent model for learning very long time dependencies.
- Rusch et al. 2022, Long expressive memory for sequence modeling.

**Questions:**

Why did you not compare with the methods in Rusch et al. 2021, Rusch et al. 2022, and Gu et al. 2021?

---

### Official Review · Reviewer_YiRU · 2023-11-01

**Soundness:** 2 fair
**Presentation:** 1 poor
**Contribution:** 1 poor
**Rating:** 3
**Confidence:** 5

**Summary:**

In this paper, the authors introduce an innovative model known as Cont-GRU, which operates in continuous time. Cont-GRU stands out for its ability to generate continuous hidden states, reset rates, and update gates. To gauge the model's performance, the authors conducted a comprehensive comparative analysis, pitting it against 17 baseline models across five real-world datasets.

**Strengths:**

1. The authors present a fully continuous GRU model.

2. To address the homeomorphic issue inherent in ODE-based methods, the authors incorporate Differential Delay Equations (DDEs).

3. The proposed model exhibits a relatively small memory footprint, making it efficient and resource-friendly.

**Weaknesses:**

1. The authors overlook several closely-related works, warranting a more comprehensive discussion of the relevant literature.

2. The introduction section lacks a clear and well-articulated explanation of the underlying intuition behind the proposed model.

3. The experimental results may benefit from greater credibility by including additional baselines that are currently absent from the analysis.

**Questions:**

1. The paper lacks a comprehensive discussion of Cont-GRU in relation to Neural CDE and Neural DDE, giving the impression that it is a straightforward amalgamation of these approaches.

2. The paper omits discussing CRU [2], a continuous recurrent model, which should be included in the related work section.

3. While the authors emphasize the limitations of Neural ODE and Neural CDE regarding the homeomorphic issue, they do not provide experimental evidence demonstrating how Cont-GRU addresses this problem.

4. Despite comparing with 17 baselines, several pertinent baseline models, such as CRU and DDE-based models, are conspicuously absent and should be included in the evaluation.

5. The authors assert that previous work represents a "half-way continuous" generalization of GRUs and argue for the necessity of fully-continuous models. However, the paper should explicitly state the limitations of fully-continuous models in certain scenarios, such as recommendation systems, where preferences can undergo abrupt changes due to external events [1].

6. The formulation of the derivative dh(t)/dt, particularly the role of A, B, and C in Equation (6), is inadequately explained in the main paper, causing confusion.

7. Although an adaptive delay factor is proposed, the paper lacks experimental validation of its effectiveness and efficiency, such as whether it reduces memory usage or accelerates the training process.

8. Memory usage and time cost should be discussed in the main paper, along with a clear explanation of why Cont-GRU is faster than NCDE.

9. Figure 1 requires improvement to enhance clarity, as the computation flow of Cont-GRU is not evident from the current depiction.

[1] Neural jump stochastic differential equations

[2] Modeling Irregular Time Series with Continuous Recurrent Units

---

### Official Review · Reviewer_JpZR · 2023-11-01

**Soundness:** 3 good
**Presentation:** 2 fair
**Contribution:** 2 fair
**Rating:** 5
**Confidence:** 3

**Summary:**

This paper proposes a new continuous GRU model for continuous time series data. It does so by defining a GRU as a delay differential equations which allows them to have a fully continuous model. The method is similar to neural controlled differential equations, with the added lag as an input.

**Strengths:**

The main method makes sense and the continuous extension of GRU is sensible. The choice of $\tau$ is convenient and I can see that it can be useful to pick this particular parameterization with adaptive solvers.

The results in Table 1, 2, 3 and 4 are really good. The method confidently beats the competitors across all tasks.

The empirical studies are a nice addition as they showcase what exactly works and what does not.

**Weaknesses:**

This is not the first continuous model in the sense of evolving the hidden state without jumps, neural CDEs are one example. It seems that the difference to neural CDEs is the network that you use, in particular you have GRU updates, and you add a single previous step from the solver.

This is also not the first neural DDE. The paper mentions two previous works but does not explain how is the proposed method different, nor is there a comparison with these methods (or explanation why it's not possible).

Although the method is presented as continuous, the main experiments are encoding fixed history and making a prediction based on the encoded vector. So the method is still encoder-decoder, therefore, you should also compare to mTAN [1] or potentially some other strong baselines. Figure 1 (c) shows that the values x(t) are continuous in time but this is only available up until the last observed data point, unless you use causal interpolation as discussed in [2]. Note that the causal cubic Hermite spline will still not allow online prediction. The method is therefore still lacking the online prediction that is possible with "jump" methods.

The implementation in code uses saving and loading previous delay values from a file which is not ideal, I can imagine some errors arrising because of that. Also, the authors should try to have a better implementation if they want wider adoption of their method.

The paper could be written better in general.

Minor:

- Limitations 1. and 2. in Introduction are not really limitations but simply design choices.

- "Eq. equation" in many places

- Algorithm 1 is not very useful, it contains too little information about the actual training procedure.

[1] Shukla & Marlin, Multi-Time Attention Networks for Irregularly Sampled Time Series (2020)

[2] Kidger et al., Neural Controlled Differential Equations for Online Prediction Tasks (2021)

**Questions:**

- GRU is surprisingly bad in Table 1 compared to your method. How do you explain this gap in performance considering that USHCN is a regularly sampled dataset?

- The baselines in Figures 5 and 6 look way off. What is the reason for such poor predictions?

- Similarly, ODERNN is very "jumpy" in predictions on USHCN in Figure 4.